# Selective Anticancer and Antimicrobial Metallodrugs Based on Gold(III) Dithiocarbamate Complexes

**DOI:** 10.3390/biomedicines9121775

**Published:** 2021-11-26

**Authors:** Elisa Abás, Diego Aguirre-Ramírez, Mariano Laguna, Laura Grasa

**Affiliations:** 1Departamento de Química Inorgánica, Instituto de Síntesis Química y Catálisis Homogénea, Universidad de Zaragoza-CSIC, Plaza S. Francisco s/n, 50009 Zaragoza, Spain; mlaguna@unizar.es; 2Departamento de Farmacología, Fisiología y Medicina Legal y Forense, Facultad de Veterinaria, Universidad de Zaragoza, Miguel Servet, 177, 50013 Zaragoza, Spain; diego23aguirre@outlook.com; 3Instituto de Investigación Sanitaria de Aragón (IIS Aragón), 50009 Zaragoza, Spain; 4Instituto Agroalimentario de Aragón -IA2, Universidad de Zaragoza-CITA, 50013 Zaragoza, Spain

**Keywords:** dithiocarbamate, cycloaurated, anticancer drugs, antimicrobial drugs, cell cycle, oxidative stress damage, apoptosis

## Abstract

New dithiocarbamate cycloaurated complexes have been synthesized and their physicochemical and in vitro antitumor properties have been evaluated. All the performed studies highlighted good transport through the blood and biodistribution, according to the balance between the properties of hydrophilicity/lipophilicity and the binding of moderate strength to the BSA protein. Furthermore, none of the complexes exhibited reduction or decomposition reactions, presenting excellent physiological stability. The in vitro cytotoxic effect was evaluated on human colon cancer cell line Caco-2/TC7, and the complexes showed great antiproliferative activity and excellent selectivity, as much less effect was detected on normal Caco-2/TC7 cells. Most of the complexes exhibit antiproliferative activity that was better than or similar to auranofin, and at least nine times better than that of cisplatin. Its action mechanism is still under discussion since no evidence of cell cycle arrest was found, but an antioxidant role was shown for some of the selective complexes. All complexes were also tested as antimicrobial drugs, exhibiting good activity towards *S. aureus* and *E. coli*. bacteria and *C. albicans* and *C. neoformans* fungi.

## 1. Introduction

Since the serendipitous discovery of cisplatin (cis-[PtCl_2_(NH_3_)_2_]) as an antitumor drug and its use in the clinic since 1978 [1], the field of investigation of metallodrugs as anticancer agents has been kept open. The main effort is focused on achieving more specific and selective drugs than cisplatin in order to hopefully erase the severe side-effects evoked by this treatment [2,3]. Platinum-based anticancer drugs have also been developed during this time, and four platinum(II) drugs are approved for use in chemotherapy [4], although cisplatin continues to be the most widely used together with the second-generation drug carboplatin [Pt(NH_3_)_2_(CBDCA-O,O′)]. Another important limitation of platinum drugs is the developing resistances shown by some kinds of tumors. For these reasons, other transition metals such as iron [5], copper [6], ruthenium [7,8,9], and gold [10,11,12], among others [13], have been considered as anticancer drugs and the results obtained in these studies are promising. The use of gold in medicine comes from its traditional use in medicine as antiarthritic agents such as gold(I) thiomalate (myocrisin), gold(I) thioglucose (solganol), and 2,3,4,6-tetra-o-acetyl-1-thio-b-D-glucopyranosato-(S)-triethylphosphine gold(I) (auranofin) (Figure 1). Some of these complexes, clinically established as antiarthritic gold(I) compounds, have shown significant antitumor properties. This dual application may be related to an anti-inflammatory effect of the gold complexes because as cancer, rheumatoid arthritis usually involves inflammatory processes [14,15,16]. 

Although gold(III) complexes seem to be better candidates to replace platinum-based drugs owing to their isostructural and isoelectronic behavior to platinum(II), the development of gold(III) complexes as alternative antitumor agents has been hindered by their poor stability under physiological conditions. This problem has been solved in the literature by the coordination of multidentate ligands to the gold center and therefore, cyclometallation is a good strategy to enhance their stability. Several cyclometalating ligands exist, where C^N is especially interesting due to the possibility of coordinating different ancillary ligands to optimize the final drug design [17,18,19,20]. In the literature, several examples of cyclometalated gold(III) complexes with (C^N), (N^N^C), or (C^N^C) ligands have been studied [21], showing promising antitumor properties. Another important family of bidentate ligands to stabilize Au^3+^ cations is dithiocarbamate ligands (DTC). Some of these complexes have shown great antiproliferative effects and stability [22,23,24]. As mentioned, the goal of the search for an alternative to platinum drugs is to avoid the resistance mechanism and the lack of selectivity. For this purpose, achieving drugs based on dithiocarbamate complexes with different action mechanisms would have a remarkable benefit against the lack of selectivity [25,26,27]. Based on these previous results, even with metals other than gold [28,29,30,31], in this work we focused on the synthesis of different cyclometalated gold(III) dithiocarbamate complexes with formula [Au(C^N)(DTC)]PF_6_ using biphenyl ligands such as 2-(*p*-tolyl)pyridinate (tpy) and 2-phenylpyridinate (ppy) as cyclometalating ligands for a potential chemotherapy application. 

On the other hand, the world is facing a crisis in treating infectious diseases, with a scarcity of new antibiotics in development to treat the growing threat of drug-resistant “superbugs” [32]. For this reason, the discovery of new antimicrobial drugs to treat some serious and life-threatening diseases is a research field of great interest. In this context, there is a large history of metallic compounds with antimicrobial properties such as cobalt [33], copper [34], and silver [35]. However, there are only a few examples of highly active gold complexes. It has been demonstrated that cyclometalated complexes have antibacterial properties, and this activity could be increased by chelating a metal to the active ligand [36]. In this work, we evaluate the antibacterial and antifungal activity of the newly synthesized cyclometalated gold(III) dithiocarbamate complexes.

## 2. Materials and Methods

### 2.1. Chemicals

^1^H and APT spectra were recorded on 400 or 300 MHz Bruker Avance spectrometers (Billerica, MA, USA) and were referenced to external TMS. Chemical shifts (δ) are given in ppm, and coupling constants are reported in Hz (Appendix A). MALDI mass spectra were measured on a Micromass Autospec spectrometer (Waters Corporation, Milford, MA, USA) in positive ion mode using DCTB (1,1-diciano-4-^t^butylphenyl-3-methylbutadiene) as the matrix. Infrared spectra (4000–250 cm^−1^) were recorded on a Perkin Elmer Spectrum 100 FTIR (far-IR) spectrometer (Zaragoza, Spain). Elemental analyses were obtained in-house using a Perkin Elmer 2400 Series II CHNS/O System elemental analyzer. The starting material HAuCl_4_ was gently donated by Torrecid S.A. Starting cyclometalated complexes [Au(C^N)Cl_2_], C^N: 2-(*p*-tolyl)pyridinate (tpy), and 2-phenylpyridinate (ppy) were obtained using a slightly modified previously described procedure [37,38]. 

### 2.2. Synthesis of Dithiocarbamate Gold(III) Cyclometalated Complex

The synthesis of dithiocarbamate gold(III) cyclometalated complexes was performed according to the protocols described before [39]. A solution of the corresponding cyclometalated complex (0.5 mmol) in acetonitrile was added to a sodium dialkyldithiocarbamate (0.5 mmol) (Alkyl: Methyl, Ethyl and Benzyl) solution in acetone dropwise to make a pale yellow–orange solution. After 30 min of stirring, a solution of potassium hexafluorophosphate (0.55 mmol) in methanol was added, enhancing the yellow color. The mixture was stirred for an additional hour and the precipitated product was filtered off and washed with distilled water (2 × 3 mL) and acetone (2 × 3 mL).

Complex [Au(tpy)(dmdtc)] PF_6_, C1: Yield, 71% ^1^H NMR (400 MHz, CDCl_3_) δ 8.91 (d, J = 5.4 Hz, 1H), 8.56–8.50 (m, 1H), 7.86 (s, 1H), 7.84 (s, 1H), 7.48–7.41 (m, 2H), 7.20 (d, J = 7.3 Hz, 1H), 3.16 (s, 3H), 3.07 (s, 3H), 2.45 (s, 3H). IR: ν (cm^−1^) = 3088, 3039, 1605, 1593, 1434, 1252, 1166, 1033, 888, 844, 821, 769, 731, 695, 416. MS^+^ (MALDI, DCTB): 485.1 (95, M-PF_6_). Anal. Calcd for C_15_H_16_AuF_6_N_2_PS_2_: C, 28.58; H, 2.56; N, 4.44; S, 10.17. Found: C, 28.63; H, 2.60; N, 4.39; S, 10.21. 

Complex [Au(ppy)(dmdtc)]PF_6_, C2: Yield, 74% ^1^H NMR (400 MHz, CDCl_3_) δ 9.84 (d, J = 5.9 Hz, 1H), 8.18–8.12 (m, 1H), 8.07 (dd, J = 8.0, 1.2 Hz, 1H), 7.92 (d, J = 7.8 Hz, 1H), 7.58–7.54 (m, 2H), 7.44–7.38 (m, 1H), 7.36 (dd, J = 10.7, 4.7 Hz, 1H), 3.51 (s, 3H), 3.48 (s, 3H). IR: ν(cm^−1^) = 3376, 3093, 1980, 1716, 1578, 1408, 832, 761, 557, 411. MS^+^ (MALDI, DCTB): 471.0 (100, (M-PF_6_)^+^). Anal. Calcd for C_14_H_14_AuF_6_N_2_PS_2_: C, 27.28; H, 2.29; N, 4.55; S, 10.41. Found: C, 27.35; H, 2.23; N, 4.52; S, 10.37.

Complex [Au(tpy)(dedtc)]PF_6_, C3: Yield, 98% ^1^H NMR (400 MHz, acetone) δ 8.82 (ddd, J = 5.8, 1.4, 0.7 Hz, 1H), 8.48 (ddd, J = 8.2, 7.4, 1.5 Hz, 1H), 8.44–8.39 (m, 1H), 7.96 (d, J = 7.9 Hz, 1H), 7.76 (ddd, J = 7.4, 5.8, 1.6 Hz, 1H), 7.39 (ddd, J = 7.9, 1.5, 0.7 Hz, 1H), 7.03 (dd, J = 1.4, 0.7 Hz, 1H), 4.07 (q, J = 7.3 Hz, 2H), 4.03 (q, J = 7.3 Hz, 2H), 2.45 (s, 3H), 1.53 (t, J = 7.3 Hz, 3H), 1.48 (t, J = 7.2 Hz, 3H). APT (101 MHz, CDCl_3_) δ 152.48, 150.22, 144.77, 144.44, 130.92, 129.68, 127.77, 126.50, 122.97, 49.64–48.12, 21.72, 12.81–12.45. IR: ν (cm^−1^) = 2978, 2249, 1605, 1562, 1465, 1285, 1037, 822, 774, 554. MS^+^ (MALDI, DCTB): 513.0 (98, (M-PF_6_)^+^). Anal. Calcd for C_17_H_20_AuF_6_N_2_PS_2_: C, 31.01; H, 3.06; N, 4.25; S, 9.74. Found: C, 31.09; H, 3.10; N, 4.29; S, 9.79.

Complex [Au(ppy)(dedtc)]PF_6_, C4: Yield, 79% ^1^H NMR (400 MHz, acetone) δ 8.64 (ddd, J = 4.8, 1.8, 0.9 Hz, 1H), 8.07–8.04 (m, 1H), 7.88 (dd, J = 5.1, 4.0, 1H), 7.85–7.77 (m, 1H), 7.67–7.63 (m, 1H), 7.29 (ddd, J = 7.4, 4.8, 1.2 Hz, 1H), 7.24–7.20 (m, 2H), 4.06 (m, 2H), 3.77 (q, J = 7.0 Hz, 2H), 1.27 (t, J = 7.1 Hz, 3H), 1.23 (m, 3H). IR: ν (cm^−1^) = 2979, 2231, 1532, 1477, 1263, 1204, 985, 829, 753, 559, 427. MS^+^ (MALDI, DCTB): 499.5 (100, (M-PF_6_)^+^). Anal. Calcd for C_16_H_18_AuF_6_N_2_PS_2_: C, 29.82; H, 2.82; N, 4.35; S, 9.95. Found: C, 29.88; H, 2.89; N, 4.31; S, 9.89.

Complex [Au(tpy)(dbdtc)]PF_6_, C5: Yield, 80% ^1^H NMR (400 MHz, acetone) δ 9.66 (d, J = 5.5 Hz, 1H), 8.15 (t, J = 8.2 Hz, 1H), 7.97 (d, J = 8.3 Hz, 1H), 7.74 (s, 1H), 7.59–7.45 (m, 2H), 7.16 (d, J = 7.7 Hz, 1H), 5.26 (s, 2H), 5.23 (s, 2H). 2.38 (s, 3H). APT (101 MHz, CDCl_3_) δ 145.70, 142.80, 132.20, 129.99, 125.40, 123.80, 120.60, 77.16, 29.86. MS^+^ (MALDI, DCTB): 485.0 (4, (M-2Ph-PF_6_)^+^), 637.0 (17, (M-PF_6_)^+^). Anal. Calcd for C_27_H_24_AuF_6_N_2_PS_2_: C, 41.44; H, 3.09; N, 3.58; S, 8.19. Found: C, 41.50; H, 3.01; N, 3.65; S, 8.23.

Complex [Au(ppy)(dbdtc)]PF_6_, C6: Yield, 76% ^1^H NMR (400 MHz, acetone) δ 8.94 (d, J = 5.5 Hz, 1H), 8.53 (m, 2H), 8.13 (d, J = 7.7 Hz, 1H), 7.82 (td, J = 5.9, 3.0 Hz, 1H), 7.59 (t, J = 7.6 Hz, 1H), 7.49–7.44 (m, 11H), 7.28 (d, J = 7.7 Hz, 1H), 5.24 (s, 2H), 5.21 (s, 2H). APT (101 MHz, CDCl_3_) δ 142.91, 132.65, 131.81, 129.38, 129.29, 129.13, 128.51, 128.21, 127.30, 125.57, 124.44, 120.92, 57.04. IR: ν (cm^−1^) = 3071, 2161, 1978, 1698, 1606, 1530, 1443, 1246, 832, 759, 733, 702, 556. MS^+^ (MALDI, DCTB): 623.0 (100, (M-PF_6_)^+^). Anal. Calcd for C_26_H_22_AuF_6_N_2_PS_2_: C, 40.63; H, 2.89; N, 3.65; S, 8.34. Found: C, 40.69; H, 2.81; N, 3.58; S, 8.29.

### 2.3. Distribution Coefficient (Log P_7.4_)

The *n*-octanol/water coefficient of the complexes was determined using the shake-flask method previously described [40]. Buffered-saline distilled water (100 mL, phosphate buffer PO_4_^3−^ = 10 mM, (NaCl = 0.15 M, pH 7.4) and *n*-octanol (100 mL) were shaken together for 72 h to allow the saturation of both phases. Approximately 1 mg of the complexes was dissolved in 5 mL of the aqueous phase, and 5 mL of the organic phase was added, mixing for 10 min. The resulting emulsion was centrifuged to separate the phases. The concentration of the compounds in each phase was determined using UV−Vis absorbance spectroscopy. Log P_7.4_ was calculated as log (Abs_organic_/Abs_aqueous_).

### 2.4. Solution Chemistry

The stability of the gold complexes was analyzed by absorption UV spectroscopy. UV–Vis absorption spectra of the complexes were recorded on a Thermo Scientific spectrophotometer (Madrid, Spain). Solutions of **C1**–**6** (5 × 10^−5^ M) in 10 mM phosphate buffer (pH = 7.4, 137 mM NaCl) were prepared from 20 mM DMSO solutions of the complexes and thereafter monitored by measuring the electronic spectra over 24 h at 37 °C. To evaluate the stability in the presence of biological reductants, all complexes were pre-solved in a minimum amount of DMSO and then diluted in 10 mM phosphate buffer (pH 7.4, 20 mM NaCl) to a final concentration of 10 µM [41]. All experiments were measure on a quartz microplate. Studies with L-glutathion reduced (GSH) and ascorbic acid (AsA) were conducted by increasing the final concentration of 0 to 50 µM (GSH) and from 0–20 µM for AsA.

### 2.5. BSA Interaction Studies

BSA was commercially available from Sigma Aldrich. A 2 mM solution of BSA was prepared in PBS (10 mM, pH = 7.4). The final concentration of the stock BSA solution was checked by UV–Vis spectra (ε_279_ nm = 43,824 M^−1^ cm^−1^). The value obtained was the real concentration of the stock solution. A 6 mM stock solution of the gold complexes in DMSO was prepared and different aliquots of the gold complex were added to a 50 μM solution of BSA in PBS and placed in a quartz cuvette of 1 cm optical path. The fluorescence spectra were recorded on a Horiba Jobin Yvon (Bensheim, Germany) fluorolog FL-3-11 spectrometer. The samples were excited at 295 nm and the emission spectra were recorded in a range from 310 to 450 nm with emission slits set to 2 nm. The fluorescence was measured for 4 min after every addition of the gold complex aliquots. The fluorescence intensities of the PBS and the gold complexes were irrelevant under the described conditions.

The data were analyzed using the Stern−Volmer equation F0/F = 1 + KSV (gold complex) = 1 + Kqτ0 (gold complex) (τ0 = lifetime of BSA in absence of the quencher) to obtain the Stern−Volmer quenching constant (K_SV_) and the quenching rate constant (Kq). The binding constant (Kb) was quantified using the modified Stern−Volmer equation: log {(F0 − F)/F} = log Kb + *n* log (gold complex) (*n* = number of binding sites) [42,43].

### 2.6. Cell Culture

This study was conducted in the human enterocyte-like cell line Caco-2/TC7 [44]. This cell line undergoes a process of spontaneous differentiation in culture that leads to the formation of a monolayer of cells, expressing the morphological and functional characteristics of the mature enterocytes. This differentiation process is growth-dependent, where the cells undergo differentiation from “undifferentiated proliferative crypt-type cells” in the exponential phase of growth, to “differentiated enterocyte-type cells” in the stationary phase [45]. The cell culture of Caco-2/TC7 cells (passages 30–50) was performed according to a previously reported method [46,47]. 

### 2.7. Cell Viability Assay

Cell survival was measured using the MTT test [48]. The assay depends on the cellular reduction of 3-(4,5-dimethylthiazol-2-yl)-2,5-diphenyltetrazolium bromide (MTT, Sigma-Aldrich, Madrid, Spain) by the mitochondrial dehydrogenase of viable cells to a blue formazan product that can be measured spectrophotometrically, as previously described [46,47]. 

### 2.8. Apoptosis Studies

For the studies of apoptosis, using the detection of phosphatidylserine in the outer layer of the plasma membrane, an Annexin V-FITC Apoptosis Detection kit (Immunostep, Salamanca, Spain) was used. After incubation with the complexes, the cells were collected and transferred to flow-cytometry tubes. A negative control was prepared with untreated cells, which was used to define the basal level of apoptotic, necrotic, or dead cells. The staining with Annexin V-FITC and 7-Aminoactinomycin D (7-AAD) was performed according to the manufacturer’s recommendations. The cells were washed twice with temperate phosphate-buffered saline (PBS) and resuspended in 200 μL of 1x Annexin-binding buffer. After that, 2.5 μL Annexin V-FITC and 2.5 μL propidium iodide (PI) were added to each 50 μL of cell suspension. After incubation for 15 min at RT in the dark, 400 μL of 1× Annexin-binding buffer were added and analyzed by flow cytometry within 1 h. The signal intensity was measured using a Gallios Flow Cytometer (Beckman Coulter, Brea, CA, USA) and analyzed using the Kaluza Analysis Software (Beckman Coulter).

### 2.9. Cell Cycle Analyses

After incubation with the complexes, the cells were collected, washed twice with PBS, fixed in 70% ice-cold ethanol, and stored at 4 °C for 24 h. After centrifugation, cells were resuspended in PBS containing PI (50 μg/mL) and RNase A (100 μg/mL). After incubation for 30 min at RT in the dark, PI-stained cells were analyzed for DNA content in a Gallios Flow Cytometer (Beckman Coulter). The red fluorescence emitted by PI was collected by a 620 nm-long pass filter as a measure of the amount of DNA-bound PI and displayed on a linear scale. Cell cycle distribution was determined on a linear scale. The percentage of cells in cycle phases was determined using the Kaluza Analysis Software (Beckman Coulter).

### 2.10. Oxidative Stress Damage

Cell homogenates needed for this assay were prepared according to a previously described method [47]. Protein oxidation was analyzed by carbonyl-level measurement as previously described [49]. Cell homogenates were incubated with the classical carbonyl reagent 2,4-dinitrophenylhydrazine (DNPH), and protein carbonylation was measured spectrophotometrically at 375 nm. The results were calculated in nmol carbonyl groups mg^−1^ protein.

### 2.11. Antimicrobial Assays

The antimicrobial assays were performed according to the protocols of the Community for Open Antimicrobial Drug Discovery (CO-ADD) [32]. **C1**–**6** were prepared in DMSO and water to a final testing concentration of 32 μg/mL and serially diluted 1:2-fold 8 times. Each sample concentration was prepared in 384-well plates, non-binding surface plate (NBS; Corning 3640) for each bacterial/fungal strain, all in duplicate (*n* = 2), and keeping the final DMSO concentration to a maximum of 0.5%. All the sample preparation was done using liquid-handling robots.

#### 2.11.1. Antibacterial Assay

The antibacterial activity of all synthesized complexes was evaluated in five different pathogenic bacteria, including methicillin-resistant *Staphylococcus aureus* (Sa), *Escherichia coli* (Ec), *Klebsiella pneumonia* (Kp), *Psedomonas aeruginosa* (Pa), and *Acinetobacter baumannii* (Ab). All bacteria were cultured in cation-adjusted Mueller Hinton Broth (CAMHB) at 37 °C overnight. A sample of each culture was then diluted 40-fold in fresh broth and incubated at 37 °C for 1.5–3 h. The resultant mid-log phase cultures were diluted (CFU/mL measured by OD_600_), then added to each well of the complexes containing plates, giving a cell density of 5 × 10^5^ CFU/mL and a total volume of 50 μL. All the plates were covered and incubated at 37 °C for 18 h without shaking. The inhibition of bacterial growth was determined by measuring the absorbance at 600 nm (OD_600_) using a Tecan M1000 Pro monochromator plate reader (Männedorf, Switzerland). The percentage of growth inhibition was calculated for each well, using the negative control (media only) and positive control (bacteria without inhibitors) on the same plate as references. The percentage of growth inhibition was calculated for each well, using the negative control (media only) and positive control (bacteria without inhibitors) on the same plate. The MIC was determined as the lowest concentration at which the growth was fully inhibited, defined by an inhibition ≥ 80%.

#### 2.11.2. Antifungal Assay

The antifungal activity of all synthesized complexes was evaluated in two different fungi, *Candida albicans* (Ca) and *Cryptococcus neoformans* (Cn). Fungi strains were cultured for 3 days on yeast extract–peptone–dextrose (YPD) agar at 30 °C. A yeast suspension of 1 × 10^6^ to 5 × 10^6^ CFU/mL (as determined by OD_530_) was prepared from five colonies. The suspension was subsequently diluted and added to each well of the complex-containing plates, giving a final cell density of fungus suspension of 2.5 × 10^3^ CFU/mL and a total volume of 50 μL. All plates were covered and incubated at 35 °C for 36 h without shaking. Growth inhibition of *C. albicans* was determined by measuring the absorbance at 630 nm (OD_630_), whereas the growth inhibition of *C. neoformans* was determined by measuring the difference in absorbance between 600 and 570 nm (OD_600–570_), after the addition of resazurin (0.001% final concentration) and incubation at 35 °C for 2 h. The absorbance was measured using a Biotek Multiflo Synergy HTX plate reader (Winooski, VT, USA). In both cases, the percentage of growth inhibition was calculated for each well, using the negative control (media only) and positive control (fungi without inhibitors) on the same plate. The MIC was determined as the lowest concentration at which the growth was fully inhibited, defined by an inhibition ≥ 80% for *C. albicans* and an inhibition ≥ 70% for *C. neoformans*. Due to a higher variance in growth and inhibition, a lower threshold was applied to the data for *C. neoformans*. 

### 2.12. Statistical Analyses

All results are expressed as means ± the standard error of the mean (SEM) of at least three independent experiments. Statistical comparisons were performed using one-way ANOVA followed by the Bonferroni post-test and differences between *p*-values < 0.05 were considered statistically significant. Statistical analyses conducted using the Prism GraphPad Program (Prism version 4.0, GraphPad Software, San Diego, CA, USA).

## 3. Results and Discussion

### 3.1. Synthesis and Characterization

The main strategy to achieve cyclometalated compounds with biphenyl ligands involves transmetalation processes with organomercurial complexes [37,50,51], but in our work great effort was made to develop new and direct synthetic routes without using any mercurial compounds. The most extended method is refluxing in MeCN/watering the precursor [AuCl_3_(CH^N)] [18,52,53]; however, this procedure is not useful for cyclometalating ligands that require high C-H bond activation. For this reason, the method developed by Grant et al. was chosen in order to obtain the desired cyclometalated precursor from an easy solventless thermal cyclometalation [38]. The dithiocarbamate derivatives were obtained by a one-pot reaction of the parent dichloro complex with the selected sodium salt of the DTC ligand. This reaction resulted in the substitution of both chlorides ligands and the formation of the cationic chelated complex. After 30 min, potassium hexafluorophosphate was added to precipitate the desired complex (Figure 1). Complexes **C1**−**6**, isolated with good yields (71%–98%), were characterized by ^1^H, ^13^C [54], mass spectrometry, IR, and elemental analysis.

^1^H-NMR data for these complexes are represented in Table 1, showing that the DTC ligand induced a high-field shift of the cyclometalated ligand signals compared to the cyclometalated precursors; although in the case of the complex with the DTC unit with the benzyl group (dbdtc) this effect was less important. In these ^1^H-NMR studies, the inequality of two different alkyl groups of the DTC moiety can be observed due to the different gold coordination environment.

The IR spectra confirmed in all cases the total displacement of chloride ligands, since no bands were found around 320 cm^−1^ corresponding to ν(Au–Cl). In contrast, a medium-intensity band above 1570 cm^−1^, attributed to the delocalized DTC NCS_2_ system [24], finally confirmed the chelation mode of the DTC ligand to the gold center. The MS-MALDI^+^ studies also pointed to the proposed structure, since a high intensity of the [M-PF_6_]^+^ peak was recorded, being the base peak of the spectrum in most of the complexes. Furthermore, there was good equivalence between the experimental and theoretical isotopic distribution for some of the synthesized complexes (Appendix A). 

### 3.2. Lipophilicity Studies

Biodistribution studies should be conducted to evaluate whether our complexes could reach their pharmacological target. The potential biodistribution of our complexes was evaluated by the measure of the partition coefficient. This parameter indicates how balanced the lipo- and hydrophilicity of the compounds are. Good equilibrium between these two properties is crucial for proper distribution inside the organism. The bloodstream is aqueous in nature, so hydrophilicity is needed for drug delivery, but too-high hydrophilicity can hinder the penetration of biological membranes. Of course, high lipophilicity behavior can hinder biomedical applications related to a good distribution profile [55]. The chosen method to measure the partition coefficient (Log P_ow_) was the shake-flask technique [40,56], with ideal values in the range of −1 ≤ Log P _ow_ ≤ 3. However, for a greater oral and intestinal absorption, a more restricted range has values around 1.35–1.80 [57]. Table 2 collects all of the obtained Log P_ow_ values, and it can be seen that, although only complex [Au(ppy)(dedtc)]PF_6_ (**C4**) satisfied this restricted range, all of them presented acceptable values of Log P_ow._ It is especially significant that in all cases, complexes with ppy ligand exhibited more lipophilic behavior than the tpy-based complexes, which was surprising because the presence of the methyl group usually increases complex lipophilicity. Something similar happened with the dbdtc moiety, since the presence of two additional phenyl rings did not seem to have a great impact on the lipophilicity/hydrophilicity balance.

### 3.3. Solution Behavior 

Due to the aqueous nature of the bloodstream, studies to understand the behavior of our complexes in such an environment were considered necessary in order to determine the stability in simulated physiological conditions, pH = 7.4 and 37 °C. By UV–Vis spectroscopy, the evolution over time of all synthesized complexes was followed for 24 h. Compounds were initially dissolved in dimethyl sulfoxide (20 µM) to reach a final concentration of 50 µM in a phosphate buffer solution (PBS) that guaranteed pH = 7.4 and contained NaCl to achieve osmolality similar to that of the human body. All complexes exhibited great stability in this medium and showed two high-intensity intramolecular intraligand bands at 214 and 256 nm, attributed to transitions located in the NCS and CSS moieties, respectively [58]. Besides, in the cases of complexes **C1**–**4**, a low-intensity broad band was detected (322 nm) (Figure 2 and Appendix A). This band was attributed to the LMCT transition, and it is very common in organometallic chemistry [59]. 

Figure 2a is represented as an example of the UV–Vis spectra collected for complex **C1**, where the absence of new bands around 550 nm allowed any decomposition process to be discarded. Furthermore, no modifications were observed in either the intensity or in the shape of the bands, which highlighted the great stability of all the complexes. Something to be mentioned is the particular case of **C5**, where a slight increase in the intensity of 328 nm band was observed during the measured time (Figure 2b). This is usually associated with the formation of Au(III)-OH bonds, increasing the water solubility and, consequently, the intensity of absorbance [60]. In biological systems, some species such as ascorbic acid (AsA) and glutathione (GSH) are present in significant concentrations, so additional studies were conducted to evaluate the susceptibility of our complexes to being reduced by these reductants. The procedure was similar to the one previously explained, but in the presence of L-glutathion reduced (GSH) and ascorbic acid (AsA). We observed that our complexes were stable with the addition of GSH over 24 h, even in a high excess of GSH to Au(III) complexes (50 vs. 16 µM) (Appendix A), since no evidence of decomposition or new bands was detected. In the specific cases of **C2** and **C3**, the band at 258 nm exhibited slight changes in intensity, but did not evolve over time. The addition of AsA did induce some significant changes in the intensity of the band centered up to 270 nm, but corresponded to the concentration of the acid rising. Again, no new bands were registered, so the reduction of Au(III) to Au(0) was ruled out (Appendix A).

### 3.4. Bovine Serum Albumin Interaction

Once in the bloodstream, drugs usually bind to the present blood proteins for delivery or excretion. According to this, the influence of these proteins on the pharmacokinetic and pharmacodynamic profiles of the complexes must be studied as part of the determination of the ADMET properties. The most abundant protein present in the plasma, and the more important one for drug delivery, is albumin. Human serum albumin (HSA) regulates osmotic blood pressure; transports Ca^2+^, Na^+^, and Zn^2+^ ions; is a carrier for many hormones, fatty acids, and exogenous substances; and is a pH regulator [61]. Structurally, the human serum albumin protein has several domains, with secondary binding sites capable of interacting and transporting metallodrugs for their biodistribution. In order to study these bindings, HSA is normally replaced by bovine analogous (BSA) as a model protein because of its similarity, commercial availability, and better fluorescence properties [62]. Although other fluorophores are present, BSA fluorescence is mainly due to two tryptophan residues [63,64].

This intrinsic fluorescence is extremely sensitive to the environment, so any binding or interaction of molecules induces changes in the BSA emission spectrum [46,47]. Consequently, interaction with BSA studies are based on monitoring the BSA luminescence in the range of 300–450 nm by adding increasing concentrations of Au(III) complexes (Figure 3a and Appendix A). The obtained fluorescence data were analyzed using the Stern–Volmer equation, which allows the Stern-Volmer quenching constant K_SV_ and the bimolecular quenching constant kq to be calculated (Figure 3b). In most of the cases studied, the Stern–Volmer representation fit a straight line, indicative of a single quenching mode, either static or dynamic. In case of drug development, static quenching is usually required to probe an interaction between the drug and the protein. By studying the bimolecular quenching constant (kq), the operated quenching can be determined since this parameter represents the efficiency of quenching or how accessible the fluorophores are to the quencher. Dynamic quenching depends on diffusion processes, so a higher kq value than the diffusion-controlled rate constant of the biomolecule in water, 10^10^ M^−1^s^−1^, may indicate a contribution of static quenching. All of our complexes showed values of kq in the order of 10^12^ M^−1^s^−1^ (Table 3), so static quenching could be established. Additionally, the modified Stern–Volmer equation permits the values of the binding constant (K_b_) for each gold complex and the number of binding sites (*n*) to be calculated. According to the K_b_ values observed in Table 3, there was a moderated interaction force between complex-BSA. These values were higher than other reported for gold(I) complexes [46,47,65], but similar to the documented interactions for potential drugs based on ruthenium or nickel [66,67]. 

Therefore, our complexes can be transported by BSA protein throughout the body, one molecule each of BSA according to the results of *n* parameter (except for **C1** or **C5**, where two molecules of gold complex could be found in the same BSA structure).

### 3.5. In Vitro Antiproliferative Activity

The in vitro cytotoxic effect was evaluated on human colon cancer cell line Caco-2/TC7. This particular cell line presents a huge advantage for toxicity studies because of its spontaneous enterocytic-like differentiation when confluence is reached. Therefore, Caco-2 cells become polarized cells expressing apical and basolateral surfaces with well-established tight junctions [68,69,70]. Thus, cells present noncarcinogenic behavior in the confluence and the cytotoxic effect of our complexes can be performed on differentiated cells to evaluate the selectivity toward tumor cells. For the antiproliferative studies, MTT assay was performed after 72 h of exposure to different concentrations of the complexes at two different stages of confluence, five days to evaluate the effect on tumor cells (Appendix A) and 15 days for normal cells (Appendix A). As can be observed in Table 4, great results were accomplished since most of our complexes showed a higher antiproliferative effect on tumor cells than cisplatin (IC_50_ = 45.60 ± 8.08 µM), auranofin (IC_50_ = 2.10 ± 0.40 µM) [27], and some other dithiocarbamates reported on [27]. The DTC group seemed to play a crucial role in the complexes to exhibit this cytotoxicity, as the cyclometalated precursors did not alter Caco-2 normal or tumor cell viability [17]. However, something to be remarkable is that our complexes exhibited excellent selectivity, 4.5–8.8-fold higher for tumor than normal cells, and especially complex **C5**, which seemed to not alter cell proliferation of normal cells up to 100 µM, with a selectivity at least 48.3 times higher for cancer cells. This fact highlights the potential application of our complexes in the chemotherapy field.

### 3.6. Apoptosis Studies

After obtaining great results from the cell viability studies, it was determined that only non-cytotoxic complexes on normal cells are good candidates as anticancer drugs, so complexes **C1**, **C5**, and **C6** were selected for the following studies. Thus, we proceeded to elucidate the action mechanism of these three complexes to induce cell death. The most convenient cell death for anticancer drugs is apoptosis, since in this way no inflammation processes occur and it is carefully controlled. The apoptotic way is normally characterized by several morphological changes in the cell membrane, by which phosphatidylserine (PS) is exposed, moving from the inner part to the outer surface of the membrane [71,72]. Thanks to this translocation, Annexin V (FITC-conjugated) can interact selectively with the PS, differentiating healthy cells from apoptotic cells. Besides, cells undergoing late apoptosis stages and necrotic cells present deep damage in the membrane that makes 7-aminoactinomycin D (7-AAD) access the nucleus. Therefore, by using flow cytometry, all different cell populations can be distinguished. In this study, no significant differences in the percentage of cells in apoptosis were detected between control and treated cell populations. According to our results, the cells did not die because of necrosis, either (Figure 4 and Appendix A).

### 3.7. Cell-Cycle Studies 

Tumor cells grow and proliferate faster than normal cells, indicating that normal cell-cycle progression has been disrupted. To stop the uncontrolled replication of tumor cells is still one of the main aims in the chemotherapy field; thus, arresting the tumor cell cycle in as many stages as possible is considered crucial. Current metallodrugs such as cisplatin or oxaliplatin show arrest of the G2/M phase [73,74], thanks to which the reproduction of tumor cells is inhibited. As is known, the geometry and structure of the complex have a great impact on the action mechanism of the drug, so there are examples of gold(III) complexes that alter the cell cycle at the G0/G1 phase [75], but also others at G2/M phase [76]. Our studies of the DNA content in Caco-2 cells stained with propidium iodide by flow cytometry showed that the major part of the cell population was in the G0/G1 phase after treatment with complexes **C1**, **C5**, or **C6**. Additionally, these complexes did not have a significant impact on the normal development of the cell cycle, as the number of cells in G2/M or S phases was similar to control (Figure 5 and Appendix A).

### 3.8. Oxidative Stress Damage

According to the literature, gold complexes act as anticancer agents by the disruption of redox homeostasis. As a result, the concentration of ROS species rises and consequently, main biomolecules, such as lipids, proteins, or even DNA, can be oxidized. However, tumor cells present an overproduction of ROS species due to their high metabolism [77]. The most common anticancer strategy is promoting a higher oxidative damage in biomolecules to trigger the apoptosis process [78]. However, there are some examples where a relationship between antioxidant and antitumor behavior has been demonstrated [79,80], even with gold(III) complexes [54,81,82]. In this study, we evaluated the oxidation levels of the carbonyl groups present in proteins by the established method of 2,4-dinitrophenylhydrazine (DNPH) [83]. The results in Figure 6 show that our three complexes significantly decreased the oxidized protein levels, indicating that our complexes behave as antioxidant agents.

Since no evidence of apoptotic (via PS exposure) or necrotic cells was detected in previous studies on cell-death mechanism, other cell-death pathways can be considered related to oxidative stress processes. Among them, autophagy is gaining attention in the chemotherapy field. Autophagy can present both negative and positive effects in cancer since autophagy inhibition can lead to tumor cell death or hyperproliferation [84]. According to previous studies, autophagy cell death is induced by stress signals, such as increased ROS, hypoxia, and mitochondrial damage [85,86]. However, based on our results, the selected gold complexes (**C1**, **C5**, and **C6**) did not promote oxidative stress damage, so our complexes do not seem to be potential autophagy promoters.

### 3.9. Antimicrobial Activity

Recently, the emergence of resistance to antimicrobial drugs and the toxicity of antimicrobials makes the development of new drugs necessary. For this purpose, all complexes were tested in five different pathogenic bacteria, including methicillin-resistant *Staphylococcus aureus*, *Escherichia coli*, *Klebsiella pneumonia*, *Psedomonas aeruginosa*, *Acinetobacter baumannii*, and two different fungi, *Candida albicans* and *Cryptococcus neoformans*. The obtained results for these studies, expressed as minimum inhibitory concentrations for microorganisms (MICs), are collected in Table 5. All the gold complexes showed antimicrobial activity, but **C1**–**4** complexes in particular showed MIC values of 0.25 µg/mL for methicillin-resistant *S. aureus*. Really low MICs, compared to those reported in other studies, of values from 2 to 15 µg/mL were obtained against this *S. aureus* strain [87,88], a bacterium that causes a range of illnesses, from minor skin infections, to life-threatening diseases such as pneumonia, meningitis, endocarditis, toxic shock syndrome, and sepsis. **C1**–**4** complexes were also shown to be effective against *E. coli*, a bacteria that can induce infections of the gastrointestinal and urinary tracts. Although the MIC values of 32 µg/mL obtained against the *E. coli* strain were significantly higher than those obtained against Sa, they can still be considered great results since other similar gold(III) complexes did not inhibit the growth of the *E. coli* strain (≤125 mg/mL) or other Gram-negative bacteria such as *K. pneumonia*, *p. aeruginosa*, or *Acinetobacter baumannii* [89,90]. Although *S. aureus* was the only Gram-positive bacteria tested, it seemed that our complexes were more effective against Gram-positive than Gram-negative bacteria. This effect was also observed in other gold complexes previously reported, [91,92]. Again, the importance of the structure on the final properties was evident, since only one CH_2_-bridge in the ancillary ligand or moving from the catecholate to dithiocarbamate group inverted the selectivity to Gram-negative [93]. The antifungal activity was also found to be acceptable, especially for complex **C6**, with really low MIC values against *C. albicans* (4 µg/mL) and *C.*
*neoformans* (≤0,25 µg/mL), and complex **C3** against *C.*
*neoformans* (4 µg/mL). The results for the fungi strains are also quite promising compared to those of other previous studies [94,95]. The SAR effect was also revealed, as **C4** and **C6** differed only in the methyl group of the pyridine ancillary ligand and their activity against *C. albicans* varied from 4 to 32 µg/mL. 

## 4. Conclusions

In this study, we report on the synthesis of new cyclometalated gold(III) complexes with dithiocarbamate moiety. The ability of dithiocarbamate complexes to access the inside of the cell was evaluated by the measure of the partition coefficient, with our complexes showing great values consequent with good biodistribution. Besides, all of them were perfectly stable in a physiological environment, according to the results of the UV–Vis studies, and no decomposition or redox process was observed. Their interaction with BSA was found to be moderate, suggesting protein-assisted transport through the blood and easy target release. All these derivatives displayed a high antiproliferative effect and excellent selectivity on Caco-2 human colon cancer cells, including one of the complexes [Au(tpy)(dbdtc)] not altering the normal cell proliferation up to 100 µM, so its selectivity was at least 48.3-fold high for cancer cells. This fact reveals the enormous impact of a single methyl group on permeability and, consequently, on the action mechanism. Despite this great antiproliferative activity, these complexes did not exhibit any tumor cell-cycle arrest like other dithiocarbamate complexes, although they acted as antioxidant agents. Therefore, additional studies are needed to elucidate the action mechanism more accurately. Additionally, antimicrobial tests were also conducted and showed promising properties, since all complexes exhibited an excellent antibacterial effect against *S. aureus* and a good effect against *E. coli*. The antifungal activity was also found to be acceptable, especially for complex **C6**, with really low MIC values against *C. albicans* and *C. neoformans*. 

## Data Availability

The authors confirm that the data supporting the findings of this study are available within the article and its Appendix A.

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
