# Peer review of "Selective Anticancer and Antimicrobial Metallodrugs Based on Gold(III) Dithiocarbamate Complexes"

_biomedicines, 2021, doi:10.3390/biomedicines9121775_

Round 1

Reviewer 1 Report

A detailed file is attached.

Author Response

Reviewer 1

In this manuscript, Grasa and coworkers presented an interesting set of new gold complexes based on the dithiocarbamate ligands.

In detail, the authors reported the synthesis and the molecular characterization of the target compounds using several techniques.

Then, the lipophilicity of the compounds was assessed as well as their stability in physiological conditions over time. Then, the interaction with BSA was evaluated by fluorescence measurements allowing to obtain quantitative data for the binding event.

Finally, these compounds were tested for their anticancer, antimicrobial and antifungal properties.

All the designed complexes exhibited high antiproliferative activity on Caco-2 human colon cancer cells, without showing relevant toxicity on the selected healthy cell counterpart thus indicating a great selectivity. Investigating the possible mechanism of action accounting for the observed anticancer activity, no apoptosis mechanism was found suggesting that another cell death pathway was probably activated.

These compounds showed also interesting antibacterial and antifungal properties, making them very attractive in multiple applications.

This manuscript is fully inserted into the promising research field of metal-based complexes; the topic is attractive and of interest for the typical readership of Biomedicines.

The experimental part is enough accurate, and the presented results are sound.

In my opinion, some check of the English language is necessary before publication since some typo errors and minor mistakes are present; also, the reference list suffers from not including many recent works and has to be improved.

Some parts have to be enriched and/or better described (see below).

This manuscript could be suitable for publication in Biomedicines once the following points are addressed.

We thank the reviewer for his positive comments on our work. We have modified the manuscript according to his suggestions regarding the English language, the addition of new bibliographic citations and the extension of some parts of the manuscript.

Major points:

  1. The use of dithiocarbamate as ligands for a metal center is not a new item. This approach has been extensively investigated especially but not only for ruthenium-based complexes. This should be discussed in the introduction adding suitable references such as:
  • Cameron, B.R. et al., Inorg Chem., 2003, 42, 4102-4108, doi: 10.1021/ic020283r;
  • Ekennia, A.C. et al., Journal of Sulfur Chemistry 2015, 36, 96-104; doi: 10.1080/17415993.2014.969731;
  • Scintilla, S. et al., J Inorg Biochem., 2016, 165, 159-169; doi: 10.1016/j.jinorgbio.2016.11.018;
  • Aldin, M.Z. et al., Journal of Organometallic Chemistry 2021, 950, 121993; doi: 10.1016/j.jorganchem.2021.121993;
  • Tan, Y.S. et al., Inorganics, 2021, 9, 60; doi: 10.3390/inorganics9080060.

As it is well mentioned by reviewer 1, we agree that dithiocarbamate ligands have been widely used in the development of metallodrugs. Precisely, the great antiproliferative effect shown by this family of complexes, impulse us to synthesize a family of different cyclometalated gold complexes with dithiocarbamate as ancillary ligand. Our aim was to enhance the stability of Au(III) by cyclometalating procedures, and therefore improve the cytotoxic effect of the dithiocarbamate moiety.

The reviewer is right that we should include other examples of metallic complexes with dithiocarbamate and no only those with gold. For this reason, we have included the works that the reviewer mentioned in the introduction (references 28-31).

  • Ekennia, A.C. et al., Journal of Sulfur Chemistry 2015, 36, 96-104; doi: 10.1080/17415993.2014.969731;
  • Scintilla, S. et al., J Inorg Biochem., 2016, 165, 159-169; doi:
  • 10.1016/j.jinorgbio.2016.11.018;
  • Aldin, M.Z. et al., Journal of Organometallic Chemistry 2021, 950, 121993; doi:
  • 10.1016/j.jorganchem.2021.121993;
  • Tan, Y.S. et al., Inorganics, 2021, 9, 60; doi: 10.3390/inorganics9080060.

  1. References related to cisplatin are not very updated. I suggest inserting in the introduction the following articles:
  • Ghosh S. Cisplatin: The first metal based anticancer drug. Bioorg Chem. 2019, 88, 102925;
  • Oun, R. et al., The side effects of platinum-based chemotherapy drugs: a review for chemists. Dalt. Trans., 2018, 47, 6645–6653.

We agree with the reviewer that the references related with cisplatin are not very updated and we have included the following articles suggested by the reviewer in the introduction (references 2-3):

  • Ghosh S. Cisplatin: The first metal based anticancer drug. Bioorg Chem. 2019, 88, 102925;
  • Oun, R. et al., The side effects of platinum-based chemotherapy drugs: a review for chemists. Dalt. Trans., 2018, 47, 6645–6653.

  1. The same for the successive references related to other transition metals such as “iron [3], copper [4], ruthenium, or gold [5-8]”

No citations for ruthenium-based complexes have been inserted.

The literature about metal-based complexes is very large. Their advantages and main features have to be highlighted, referring to recently published reviews.

  • Khoury A. et al., Recent Advances in Platinum-Based Chemotherapeutics That Exhibit Inhibitory and Targeted Mechanisms of Action J Inorg Biochem. 2020 207:111070. doi: 10.1016/j.jinorgbio.2020.111070.
  • Omondi et al. Review of comparative studies of cytotoxic activities of Pt(II), Pd(II), Ru(II)/(III) and Au(III) complexes, their kinetics of ligand substitution reactions and DNA/BSA interactions. Inorganica Chimica Acta, 2020, 512, 119883.
  • Riccardi C. et al., Anticancer Ruthenium(III) Complexes and Ru(III)-Containing Nanoformulations: An Update on the Mechanism of Action and Biological Activity. Pharmaceuticals 2019; 2(4):146. doi: 10.3390/ph12040146.
  • Murray et al., Recent Progress in the Development of Organometallics for the Treatment of Cancer Curr Opin Chem Biol. 2019 56:28-34. doi: 10.1016/j.cbpa.2019.11.001.

This is a selected list!

References for ruthenium-based complexes have been included in the manuscript, although it is true that ruthenium and gold references are putting all together. We apologize for the mistake and we have included the citations separated in the introduction: “For these reasons, other transition metals such as iron [5], copper [6], ruthenium [7-9], or gold [10-12]

Most of the presented references are from 4 years ago, so in our opinion they are not so old-dated. We agree that there are others more recent studies, thus we have included the following suggested citations (7, 12, 13, in the Introduction):

  • Omondi et al. Review of comparative studies of cytotoxic activities of Pt(II), Pd(II), Ru(II)/(III) and Au(III) complexes, their kinetics of ligand substitution reactions and DNA/BSA interactions. Inorganica Chimica Acta, 2020, 512, 119883.
  • Riccardi C. et al., Anticancer Ruthenium(III) Complexes and Ru(III)-Containing Nanoformulations: An Update on the Mechanism of Action and Biological Activity. Pharmaceuticals 2019; 2(4):146. doi: 10.3390/ph12040146.
  • Murray et al., Recent Progress in the Development of Organometallics for the Treatment of Cancer Curr Opin Chem Biol. 2019 56:28-34. doi: 10.1016/j.cbpa.2019.11.001.

  1. “The main strategy to achieve such cyclometalated compounds with biphenyl ligands involves transmetallation processes with organomercurial complexes [23-25], but in our work a great effort was made to develop new and direct synthetic routes without using any mercurial compounds. The most extended method is reflux in MeCN/water the precursor [AuCl3(CH^N)] [13, 26, 27], however this procedure is not useful for cyclometallating ligands that require high C-H bond activation. For this reason, the method developed by G.J. Grant et al. was chosen in order to obtain the desired cyclometalated pre-cursor from an easy solventless thermal cyclometallation [28].”

In my opinion, this part should be moved to results and discussion because it describes the design of your ligands and the synthetic methods exploited for cyclometallation.

We agree with the reviewer and consequently, we have removed this paragraph in the “Introduction” and we have moved it to “Results and discussion”, in 3.1. Synthesis and Characterization section.

  1. In the “Materials and methods” section, the description of 13C NMR spectra is completely missing.

We have tried to add the 13C spectra of the synthesized complexes but with only 7 days of time to review, we have not been able to obtain adequate results for their incorporation.

  1. “Solution chemistry”: in this section, we first reported to use a phosphate buffer (pH = 7.4) to dissolve the complexes and monitor their UV-vis absorption spectra and then use a 10 mM phosphate buffer (pH 7.4, 20 mM NaCl) for the stability studies in the presence of biological reductants. Are the buffer solution used the same? If not, why?

In addition, in both experiments, you started from complexes dissolved in DMSO. Have you considered the contribution of this solvent subtracting adequately it from the recorded spectra?

The buffer solutions used for these experiments are not the same since, according other reported works, the concentration of NaCl in the phosphate buffer must be 20 mM for stability in the presence of AsA or GSH. However, the standard and commercial PBS (x1) usually presents a 137 mM NaCl concentration. The citation 41 has been included in the “Materials and Methods”, in 2.4. Solution Chemistry section.

In both experiments, we started from complexes dissolved in DMSO. Of course, the first step of the protocol was to make an adequate blank, and the used blank was the same volume of DMSO and phosphate buffer in the same volume as the samples, the only different is that in the case of sample the volume of DMSO is replaced by gold complex in DMSO.

  1. “Furthermore, there was a good equivalence between the experimental and theoretical isotopic distribution.”

How do you determine the theoretical isotopic distribution for these compounds? Please, provide details in the experimental section. Also, it could be very useful to show for at least one compound the comparison of the theoretical and experimental MS spectrum obtained. Please, insert it.

The software used to predict the theoretical isotopic distribution of the complexes was Bruker Compass Data Analysis 4.0. This tool allow us to simulate the corresponding distribution of the indicated fragment. According to the suggestion of the reviewer, we have inserted the comparison between the theoretical and experimental MS spectrum obtained for complexes C2, C3 and C6 in the supporting information (Figure S1).

  1. “Complexes C1-4 a low-intensity broad band was detected (322 nm), probably due to LMCT transition.”

This is a behaviour typically found in metal complexes, such as those based on platinum and ruthenium ions. See for example https://doi.org/10.1021/acsbiomaterials.7b00547.

We certainly know about this common behavior in metal complexes but we wanted to be cautious with our conclusions. As the reviewer suggests, we have rewritten this sentence in the section 3.3. Solution Behaviour and we have included the citation with the number 59: “Besides, in cases of complexes C1-4 a low-intensity broad band was detected (322 nm) (Figure 2, Figure S2). This band is attributed to LMCT transition, and it is something very common in the organometallic chemistry [59].”

  1. “Furthermore, no significant modifications were observed not only in the intensity but also in the shape of the bands, facts that highlighted a great stability of all complexes.”

In my opinion, a modification of the intensity of the main UV bands occurs. These variations are not so relevant but they are. I suggest changing this discussion.

We totally agree with the reviewer that the modifications of the intensity are irrelevant, for this reason we indicated that they are not significant. Perhaps the word significant is not correct, and this sentence has been rewritten in the section 3.3. Solution behavior.

  1. “We observed that our complexes are stable by the addition of GSH over 24 h even in a high excess of GSH to Au(III) complexes (50 vs. 16 μM) (Figure S2).”

Also in this case, a slight reduction of the UV intensity occurs, even if is not so relevant as in the case of the AsA addition. However, especially for C2 and C3 compounds, slight changes are visible and thus have to be described.

Following the instructions of the reviewer, we have modified this section (page 9) and we complemented with a brief discussion of the C2 and C3 cases.

“We observed that our complexes are stable by the addition of GSH over 24 h, even in a high excess of GSH to Au(III) complexes (50 vs. 16 µM) (Figure S3), since no evidence of decomposition or new bands were detected. In the specific cases of C2 and C3, the band at 258 nm exhibited slight changes in their intensities, but they did not evolve with time. The addition of AsA did induce some important changes in the intensity of the band centered up to 270 nm, but correspond to the concentration rising of the acids. Again, no new bands were registered, so the reduction of Au(III) to Au(0) was ruled out (Figure S4).”

  1. Fluorescence-based interaction studies with the BSA are not described in the experimental part. Please, add this information.

We apologize for the mistake, and of course the corresponding BSA studies procedure was added as the needed citations (page 4).

In addition, regarding these experiments, Figures 3 and S4 should be modified with a more detailed legend, which associates each color to the concentration of gold complex used. Also, starting from the fixed concentration used for BSA (which is in the present manuscript form an interrogative point for the reader), it could be interesting to know the molar ratio of BSA/Au(III) complexes at each point of the titrations.

In turn, Figure 3b is very poorly resolved. Please, improve the quality of this picture.

Figure 3 and all the images collected in Figure S4 have been modified in order to incorporate a more detailed legend, and also an extra resolution for a better quality.

The started concentration of BSA is 50 µM, it is now indicated in the Experimental Section. Also, the corresponding ratio, as the reviewer demanded, is specified in all spectra.

However, acquired fluorescence spectra have to be properly discussed: a quenching phenomenon is observed on increasing the concentration of each Au(III) complex. This is a behaviour typically found when a metal complex interacts with BSA as previously reported for other gold complexes. See for example this recent paper: https://doi.org/10.3390/biom11101449. This manuscript is very interesting since the interaction with BSA was also studied by CD spectroscopy, allowing detecting and quantifing a reduction of the α-helix content in the BSA structure. These kinds of experiments could give precious information on the here proposed Au(III) complexes and thus I strongly suggest performing them.

We totally agree with the reviewer  that the mentioned research is really interesting, and we are aware of the CD studies to further evaluate BSA-metal complex bindings. However, we have not access to a CD equipment, although we are working in the search of collaborations or CD equipment for futures studies.

  1. How do you explain that C1 and C5 analogues bind BSA in a 2:1 complex/protein ratio?

We are still looking for an explanation to understand this shocking results, not only for these two complexes but also to explain why dmdtc and dbdtc-based complexes exhibited higher values of n factor than dedct complexes. We think that probably, due to the different size and chemistry-related of the dithiocarbamate ligand, the operating interaction is not the same and maybe, C1 (smaller than C5) can insert in a pocket of BSA, whereas C5 may use its Ph-ligand for this bond.

We hope that in the future can develop some theoretical or modeling studies, but for the moment we have not enough experience in this field.

  1. I think that the studies on antiproliferative activity of the here designed Au(III) complexes should be expanded to other well-established models of cancer cells. Antimicrobial activity was for example assessed on different bacteria, while in the case of antiproliferative activity only one cell type was examined along with its healthy counterpart. Other cancer cells, with different histological origin, have to be taken in consideration.

Thank you for the suggestion, and of course it will be taken in consideration for future studies. However, in this research, we mainly study potential metallodrugs against Caco-2 cells.

  1. Why did you select only compounds C1, C5 and C6 for apoptosis, cell cycle and oxidative stress damage studies? Please, explain this choice.

Our research is focused on the improvement of the actual used metallodrug, and therefore our main goal is to achieve selective complexes towards cancer cells. For this reason, we only keep working with those complexes which exhibit this selectivity upon 20 µM on non-tumor Caco-2 cells. 

  1. The evaluation of mechanism of action underlined that no apoptosis occurs after gold complexes treatment. This result suggested that probably other cell death pathways are involved in the observed anticancer effect such as for example autophagia, which is a mechanism often activate by metal-based complexes (see for example doi: 10.1038/s41598-019-43411-3). This point should be better discussed.

Following the suggestion of the reviewer, we have enclosed a brief discussion about the inducement of autophagy mechanism in the oxidative stress studies section (page 12).

“Since no evidence of apoptotic (via PS exposure) or necrotic cells were detected in previous studies on cell death mechanism, other cell death pathways can be considered related to oxidative stress processes. Among them, autophagy is gaining attention in chemotherapy field. Autophagy can present both negative and positive effects in cancer since autophagy inhibition can lead to tumor cell death or hyperproliferation [84]. According to previous studies, autophagy cell death is induced by stress signals, such as increased ROS, hypoxia, and mitochondrial damage [85,86]. However, based on our results, the selected gold complexes (C1, C5 and C6) did not promote oxidative stress damage, on the contrary, so our complexes do not seem to be potential autophagy promoters”.

A list of minor points to be fixed before publication is here enclosed:

  1. Page 1, Introduction: “Since the serendipitous discover of cisplatin…”

Did you mean “discovery”?

  1. Page 1, Introduction: “…although cisplatin continues being the most widely used…”

It could be better: “…although cisplatin continues to be the most widely used…”

  1. Page 1, Introduction: “…is the developing resistances showed by some kinds of tumors.”

I suggest changing with: “…is the developing resistances shown by some kinds of tumors.”

  1. Page 2, Introduction: “Although gold(III) complexes seem to be better candidates to replace platinum-based drugs owing to its isostructural…”

It should be: “… to their isostructural….”

  1. Page 2, Introduction: “For this propose…”

Did you mean “purpose”?

  1. Page 6, 3.1. Synthesis and Characterization: “…to precipitate the desire complex…”

Please, change “desire” with “desired”.

  1. Page 7, 3.1. Synthesis and Characterization: “In acetone-d6.”

6 should be subscript.

  1. Page 7, 3.3. Solution Behavior: “phosphate buffer solution (PB) that guarantee…”

It should be phosphate buffer solution (PBS) that guarantees…. Please, specify the buffer composition.

  1. Page 8, 3.3. Solution Behavior: “Caption of Figure 2”

Please, insert the names given to [Au(tpy)(dmdtc)]PF6 and [Au(tpy)(dbdtc)]PF6 complexes, which should be C1 and C5 respectively (see also the following point).

  1. Page 8, 3.3. Solution Behavior: “Figure 2a is represented as an example of the UV/vis spectra collected for complex C2”

Please, check and correct the caption of the Figure and the name of the complex through the text. Complex C2 is [Au(ppy)(dmdtc)]PF6 but in the caption of the Figure you refer to Au(tpy)(dmdtc)]PF6, which is C1. Please, clarify this point.

  1. Page 8, 3.3. Solution Behavior: “…during the measure time…”

It should be “the measured time”.

  1. Page 8, 3.4. Bovine Serum Albumin Interaction: “…must to be studied…”

The word to is usually unnecessary after the modal verb must. Consider removing it.

  1. Page 9, 3.4. Bovine Serum Albumin Interaction: “which allows to calculate…”

Allows calculating…sounds better.

We would like to thank the reviewer for the detailed list of English mistakes, all of them have been taken into account and the entire manuscript has been checked to ensure an adequate grammar English.

Reviewer 2 Report

Accept

Author Response

We would like to thank the reviewer for considering our paper is ready to be publish.

Reviewer 3 Report

The manuscript describes the cytotoxicity and antimicrobial activities of some synthesised metal complexes. In general, the manuscript lack specificity as two different bioactivities have been screened. While it is not wrong to screen for two different activities. But the quality of the presentation of data is low. It is advisable to restructure the whole manuscript prior to submission. The presentation of the data is not consistent. For example, C1-5, 1-5 etc.

Author Response

Reviewer 3

The manuscript describes the cytotoxicity and antimicrobial activities of some synthesised metal complexes. In general, the manuscript lack specificity as two different bioactivities have been screened. While it is not wrong to screen for two different activities. But the quality of the presentation of data is low. It is advisable to restructure the whole manuscript prior to submission. The presentation of the data is not consistent. For example, C1-5, 1-5 etc.

We thank the reviewer for the comments. Some parts of the manuscript have been relocated. Hence, for a better consistent presentation, every time that we have mentioned several complexes is written like C1-6. In addition, we have greatly improved the work with the specific comments provided by the reviewer 1, in aspects such as the English revision, improving the quality of the figures, improving the introduction, methodology and discussion of the results, and adding new figures.

Although the reviewer mention a lack of specificity, we think that we used the same colon cancer cells to perform all the studies to finally understand the whole mechanism of our complexes.

We consider to carry out antimicrobial studies especially interesting, since in the pharmaceutical industry, drugs that can be used for various diseases.

Round 2

Reviewer 1 Report

In this revised version of their manuscript, Grasa and coworkers sensibly improved the presentation of data and overall discussion, taking into account all the Reviewer's suggestions.

In the present form, the paper is better focused and centered in the background context.

So, I suggest its publication in Biomedicines, even if some of the raised points were not completely addressed. For example:

Major point 5. In the “Materials and methods” section, the description of 13C NMR spectra is completely missing.

We have tried to add the 13C spectra of the synthesized complexes but with only 7 days of time to review, we have not been able to obtain adequate results for their incorporation.

I asked for these spectra because in the experimental section you declared to have performed them. If it is not true, please remove the statement in the 2.1. paragraph entitled “Chemicals”.

Major point 6. “Solution chemistry”: in this section, you first reported to use a phosphate buffer (pH = 7.4) to dissolve the complexes and monitor their UV-vis absorption spectra and then use a 10 mM phosphate buffer (pH 7.4, 20 mM NaCl) for the stability studies in the presence of biological reductants. Are the buffer solutions used the same? If not, why?

The buffer solutions used for these experiments are not the same since, according other reported works, the concentration of NaCl in the phosphate buffer must be 20 mM for stability in the presence of AsA or GSH. However, the standard and commercial PBS (x1) usually presents a 137 mM NaCl concentration. The citation 41 has been included in the “Materials and Methods”, in 2.4. Solution Chemistry section.

If I correctly understand, the only difference between the selected buffer solutions is the addition of 20 mM NaCl to improve the stability in the presence of the selected reductants? Is it true?

My doubts derive from the fact that “phosphate buffer” has no meaning. You should specify the prevalent ions in solution, so if it is a sodium, a potassium or other buffer solutions and the concentration that I suppose to be 10 mM but it is not clear.

Major point 7. How do you determine the theoretical isotopic distribution for these compounds? Please, provide details in the experimental section. Also, it could be very useful to show for at least one compound the comparison of the theoretical and experimental MS spectrum obtained. Please, insert it.

The software used to predict the theoretical isotopic distribution of the complexes was Bruker Compass Data Analysis 4.0. This tool allow us to simulate the corresponding distribution of the indicated fragment. According to the suggestion of the reviewer, we have inserted the comparison between the theoretical and experimental MS spectrum obtained for complexes C2, C3 and C6 in the supporting information (Figure S1).

Thank you for the addition of the Figure in the supporting information, but also the tool used should be reported or in the experimental part or added as information to the caption of Figure S1.

Author Response

Reviewer 1

Major point 5. In the “Materials and methods” section, the description of 13C NMR spectra is completely missing.

We have tried to add the 13C spectra of the synthesized complexes but with only 7 days of time to review, we have not been able to obtain adequate results for their incorporation.

I asked for these spectra because in the experimental section you declared to have performed them. If it is not true, please remove the statement in the 2.1. paragraph entitled “Chemicals”.

Thanks for the comment. We apologize again for the misunderstanding in this point. In fact, the APT 13C spectra of complexes C3, C5 and C6 are described in the manuscript in the section 2.2. “Synthesis of dithiocarbamate gold(III) cyclometalated complexes” (marked now in red in the manuscript). That´s why we keep the statement in the paragraph 2.1. Chemicals “ 1H and 13C NMR spectra were recorded on 400 or 300 MHz Bruker Avance spectrometers and were referenced to external TMS”. However, we have removed the part of  “or 85% H3PO4 (31P)” because we do not present 31P spectra.

Major point 6. “Solution chemistry”: in this section, you first reported to use a phosphate buffer (pH = 7.4) to dissolve the complexes and monitor their UV-vis absorption spectra and then use a 10 mM phosphate buffer (pH 7.4, 20 mM NaCl) for the stability studies in the presence of biological reductants. Are the buffer solutions used the same? If not, why?

The buffer solutions used for these experiments are not the same since, according other reported works, the concentration of NaCl in the phosphate buffer must be 20 mM for stability in the presence of AsA or GSH. However, the standard and commercial PBS (x1) usually presents a 137 mM NaCl concentration. The citation 41 has been included in the “Materials and Methods”, in 2.4. Solution Chemistry section.

If I correctly understand, the only difference between the selected buffer solutions is the addition of 20 mM NaCl to improve the stability in the presence of the selected reductants? Is it true?

My doubts derive from the fact that “phosphate buffer” has no meaning. You should specify the prevalent ions in solution, so if it is a sodium, a potassium or other buffer solutions and the concentration that I suppose to be 10 mM but it is not clear.

Thanks for the comment. We apologize again for the misunderstanding in this point. The PBS buffers used are different only in the NaCl concentrations. We used a 10 mM phosphate buffer (pH = 7.4, NaCl 137 mM) to dissolve the complexes and monitor their UV-vis absorption spectra. We used a 10 mM phosphate buffer (pH = 7.4, NaCl 20 mM)  for the stability studies in the presence of biological reductants. We have modified the section 2.4 to clarify this point as follows:

“The stability of the gold complexes has been analyzed by absorption UV spectroscopy. UV-Vis absorption spectra of the complexes were recorded on a Thermo Scientific spectrophotometer. Solutions of C1–6 (5·10−5 M) in 10 mM phosphate buffer (pH = 7.4, 137 mM NaCl) were prepared from 20 mM DMSO solutions of the complexes and thereafter monitored by measuring the electronic spectra over 24 h at 37 °C. To evaluate the stability in the presence of biological reductants, all complexes were pre-solved in a minimum amount of DMSO, and then diluted in 10 mM phosphate buffer (pH 7.4, 20 mM NaCl) to a final concentration of 10 µM [41].“

Major point 7. How do you determine the theoretical isotopic distribution for these compounds? Please, provide details in the experimental section. Also, it could be very useful to show for at least one compound the comparison of the theoretical and experimental MS spectrum obtained. Please, insert it.

The software used to predict the theoretical isotopic distribution of the complexes was Bruker Compass Data Analysis 4.0. This tool allow us to simulate the corresponding distribution of the indicated fragment. According to the suggestion of the reviewer, we have inserted the comparison between the theoretical and experimental MS spectrum obtained for complexes C2, C3 and C6 in the supporting information (Figure S1).

Thank you for the addition of the Figure in the supporting information, but also the tool used should be reported or in the experimental part or added as information to the caption of Figure S1.

Thanks for the comment. We have included the tool used in the caption of Figure S1:

Figure S1. Comparison of the theoretical and experimental MS spectra of complexes C2, C3 and C6 (from left to right). The software used to predict the theoretical isotopic distribution of the complexes was Bruker Compass Data Analysis 4.0.

Reviewer 3 Report

The title is misleading. The author claimed that the synthesised compounds are potent for anticancer and antimicrobial activities. Referring to table 4, the concentration of the best compound from the series is more than 1 uM. In addition, the best MIC data for the compound is about 4 uM, which is beyond for it as a potent compound or drug. Please revise the title accordingly. 

Please check the style of writting extensively. 

Author Response

Reviewer 2

The title is misleading. The author claimed that the synthesised compounds are potent for anticancer and antimicrobial activities. Referring to table 4, the concentration of the best compound from the series is more than 1 uM. In addition, the best MIC data for the compound is about 4 uM, which is beyond for it as a potent compound or drug. Please revise the title accordingly. 

Thanks for the comment. Based on the reviewer's suggestion, we have removed the word “Potent” from the title of the work.

Please check the style of writting extensively. 

Thanks for the comment. We have revised the English style and writting with the Trinka tool, a grammar and language checker for academic works. The changes have been incorporated into the manuscript with the change control.